# ADVERSARIAL IMITATION ATTACK

## ABSTRACT

Deep learning models are known to be vulnerable to adversarial examples. A practical adversarial attack should require as little as possible knowledge of attacked models. Current substitute attacks need pre-trained models to generate adversarial examples and their attack success rates heavily rely on the transferability of adversarial examples. Current score-based and decision-based attacks require lots of queries for the attacked models. In this study, we propose a novel adversarial imitation attack. First, it produces a replica of the attacked model by a two-player game like the generative adversarial networks (GANs). The objective of the generative model is to generate examples which lead the imitation model returning different outputs with the attacked model. The objective of the imitation model is to output the same labels with the attacked model under the same inputs. Then, the adversarial examples generated by the imitation model are utilized to fool the attacked model. Compared with the current substitute attacks, imitation attack can use less training data to produce a replica of the attacked model and improve the transferability of adversarial examples. Experiments demonstrate that our imitation attack requires less training data than the black-box substitute attacks, but achieves an attack success rate close to the white-box attack on unseen data with no query.

## 1 INTRODUCTION

Deep neural networks are often vulnerable to imperceptible perturbations of their inputs, causing incorrect predictions (Szegedy et al., 2014). Studies on adversarial examples developed attacks and defenses to assess and increase the robustness of models, respectively. Adversarial attacks include white-box attacks, where the attack method has full access to models, and black-box attacks, where the attacks do not need knowledge of models structures and weights.

White-box attacks need training data and the gradient information of models, such as FGSM (Fast Gradient Sign Method) (Goodfellow et al., 2015), BIM (Basic Iterative Method) (Kurakin et al., 2017a) and JSMA (Jacobian-based Saliency Map Attack) (Papernot et al., 2016b). However, the gradient information of attacked models is hard to access, the white-box attack is not practical in real-world tasks. Literature shows adversarial examples have transferability property and they can affect different models, even the models have different architectures (Szegedy et al., 2014; Papernot et al., 2016a; Liu et al., 2017). Such a phenomenon is closely related to linearity and over-fitting of models (Szegedy et al., 2014; Hendrycks & Gimpel, 2017; Goodfellow et al., 2015; Tramèr et al., 2018). Therefore, substitute attacks are proposed to attack models without the gradient information. Substitute black-box attacks utilize pre-trained models to generate adversarial examples and apply these examples to attacked models. Their attack success rates rely on the transferability of adversarial examples and are often lower than that of white-box attacks. Black-box score-based attacks (Chen et al., 2017; Ilyas et al., 2018a;b) do not need pre-trained models, they access the output probabilities of the attacked model to generate adversarial examples iteratively. Black-box decision-based attacks (Brendel et al., 2017; Cheng et al., 2018; Chen et al., 2019) require less information than the score-based attacks. They utilize hard labels of the attacked model to generate adversarial examples.

Adversarial attacks need knowledge of models. However, a practical attack method should require as little as possible knowledge of attacked models, which include training data and procedure, models weights and architectures, output probabilities and hard labels (Athalye et al., 2018). The disadvantage of current substitute black-box attacks is that they need pre-trained substitute models trained by

the same dataset with attacked model $T$ (Hendrycks & Gimpel, 2017; Goodfellow et al., 2015; Kurakin et al., 2017a) or a number of images to imitate the outputs of $T$ to produce substitute networks (Papernot et al., 2017). Actually, the prerequisites of these attacks are hard to obtain in real-world tasks. The substitute models trained by limited images hardly generate adversarial examples with well transferability. The disadvantage of current decision-based and score-based black-box attacks is that every adversarial example is synthesized by numerous queries.

Hence, developing a practical attack mechanism is necessary. In this paper, we propose an adversarial imitation training, which is a special two-player game. The game has a generative model $G$ and a imitation model $D$. The $G$ is designed to produce examples to make the predicted label of the attacked model $T$ and $D$ different, while the imitation model $D$ fights for outputting the same label with $T$. The proposed imitation training needs much less training data than the $T$ and does not need the labels of these data, and the data do not need to coincide with the training data. Then, the adversarial examples generated by $D$ are utilized to fool the $T$ like substitute attacks. We call this new attack mechanism as adversarial imitation attack. Compared with current substitute attacks, our adversarial imitation attack requires less training data. Score-based and decision-based attacks need a lot of queries to generate each adversarial attack. The similarity between the proposed method and current score-based and decision-based attacks is that adversarial imitation attack also needs to obtain a lot of queries in the training stage. The difference between these two kinds of attack is our method do not need any additional queries in the test stage like other substitute attacks. Experiments show that our proposed method achieves state-of-the-art performance compared with current substitute attacks and decision-based attack. We summarize our main contributions as follows:

- The proposed new attack mechanism needs less training data of attacked models than current substitute attacks, but achieves an attack success rate close to the white-box attacks.

- The proposed new attack mechanism requires the same information of attacked models with decision attacks on the training stage, but is query-independent on the testing stage.

## 2 RELATED WORK

**Adversarial Scenes**  Adversarial attacks happen in two scenes, namely the white-box and the black-box settings. In the white-box settings, the attack method has complete access to attacked models, such as models internal, training strategy and data. While in the black-box settings, the attack method has little knowledge of attacked models. The black-box attack utilizes the transferability property of adversarial examples, only needs the labeled training data, but its attack success rate is often lower than that of the white-box attack method if attacked models have no defense. Actually, attack methods requiring lots of prior knowledge of attacked models are difficult to apply in practical applications (Athalye et al., 2018).

**Adversarial Attacks**  Several methods for generating adversarial examples were proposed. Goodfellow et al. (2015) proposed a one-step attack called FGSM. On the basis of the FGSM, Kurakin et al. (2017a) came up with BIM, an iterative optimization-based attack. Another iterative attack called DeepFool (Moosavi-Dezfooli et al., 2016) aims to find an adversarial example that would cross the decision boundary. Carlini & Wagner (2017b) provided a stronger attack by simultaneously minimizing the perturbation and the $L_F$ norm of the perturbation. Rony et al. (2018) generated adversarial examples through decoupling the direction and the norm of the perturbation, which is also constrained by the $L_F$ norm. Liu et al. (2017) showed that targeted adversarial examples hardly have transferability, they proposed ensemble-based methods to generate adversarial examples having stronger transferability. Papernot et al. (2017) proposed a practical black-box attack which accesses the hard label to train substitute models. For score-based attacks, Chen et al. (2017) proposed the zeroth-order based attack (ZOO) which uses gradient estimates to attack a black-box model. Ilyas et al. (2018b) improves the way to estimate gradients. Guo et al. (2019) proposed a simple black-box score-based attack on DCT space. For decision-based attacks, Brendel et al. (2017) first proposed decision-based attacks which do not rely on gradients. Cheng et al. (2018) and Chen et al. (2019) improve the query efficiency of the decision-based attack.

**Adversarial Defenses**  To increase the robustness of models, methods for defending against adversarial attacks are being proposed. Adversarial training (Szegedy et al., 2014; Madry et al., 2018;

Kurakin et al., 2017b; Tramèr et al., 2018) can be considered as a kind of data augmentation. It applies adversarial examples to the training data, resulting in a robust model against adversarial attacks. Defenses based on gradient masking (Tramèr et al., 2018; Dhillon et al., 2018) provide robustness against optimization-based attacks. Random transformation (Kurakin et al., 2017a; Meng & Chen, 2017; Xie et al., 2018) on inputs of models hide the gradient information, eliminate the perturbation. Buckman et al. (2018) proposed thermometer encoding based on one-hot encoding, it applied a nonlinear transformation to inputs of models, aiming to reduce the linearity of the model. However, most defenses above are still unsafe against some attacks (Carlini & Wagner, 2017a; He et al., 2017). Especially Athalye et al. (2018) showed that defenses based on gradient masking actually are unsafe against attacks. Instead, some researchers focus on detecting adversarial examples. Some use a neural network (Gong et al., 2017; Grosse et al., 2017; Metzen et al., 2017) to distinguish between adversarial examples and clean examples. Some achieve statistical properties (Bhagoji et al., 2017; Hendrycks & Gimpel, 2017; Feinman et al., 2017; Ma et al., 2018; Papernot & McDaniel, 2018) to detect adversarial examples.

## 3 IMITATION ATTACK

In this section, we introduce the definition of adversarial examples and then propose a new attack mechanism based on adversarial imitation training.

### 3.1 ADVERSARIAL EXAMPLES

$\mathbf{X}$ refers to the samples from the input space of the attacked model $T$, $y_{true}$ refers to the true labels of the samples $\mathbf{X}$. $T(y|\mathbf{X}, \theta)$ is the attacked model parameterized by $\theta$. For a non-targeted attack, the objective of the adversarial attack can be formulated as:

$$\min_{\epsilon} \|\epsilon\| \quad \text{subject to} \quad \underset{y_i}{\text{argmax}} \ T(y_i|\overline{\mathbf{X}} = \mathbf{X} + \epsilon, \theta) \neq y_{true}$$
$$\text{and} \quad \|\epsilon\| \leq r, \tag{1}$$

where the $\epsilon$ and $r$ are perturbation of the sample and upper bound of the perturbation, respectively. To guarantee that $\epsilon$ is imperceptible, $r$ is set to a small value in applications. $\overline{\mathbf{X}} = \mathbf{X} + \epsilon$ are the adversarial examples which can fool the attacked model $T$. $\theta$ refers to the parameters of the model $T$.

For white-box attacks, they obtain gradient information of $T$ to generate adversarial examples, and attack the $T$ directly. For substitute attacks, they generate adversarial examples from a substitute model $\widehat{T}$, and transfer the examples to attack the $T$. The key point of a successful attack is the transferability of the adversarial examples.

To improve the transferability of adversarial examples and avoid output query, we utilize an imitation network to imitate the characteristics of the $T$ by accessing its output labels to improve the transferability of adversarial examples, which are generated by the imitation network. After the adversarial imitation training, adversarial examples generated by imitation network do not need additional query. In the next subsection, we introduce the proposed adversarial imitation training and imitation attack.

### 3.2 ADVERSARIAL IMITATION TRAINING

Inspired by the generative adversarial network (GAN) (Goodfellow et al., 2014), we use the adversarial framework to copy the attacked model. We propose a two-player game based adversarial imitation training to replicate the information of attacked model $T$, which is shown in Figure 1. To learn the characteristics of $T$, we define an imitation network $D$, and train it using disturbed input $\widehat{\mathbf{X}} = G(\mathbf{X}) + \mathbf{X}$ and the corresponding output label $y_T(\widehat{\mathbf{X}})$ of attacked model. $\mathbf{X}$ denotes training samples here. The role of $G$ is to create new samples that $y_T(\widehat{\mathbf{X}}) \neq y_D(\widehat{\mathbf{X}})$. Thus, $D$, $G$ and $T$ play a special two-player game. To simplify the expression but without loss of generality, we just analyze the case of binary classification. The value function of players can be presented as:

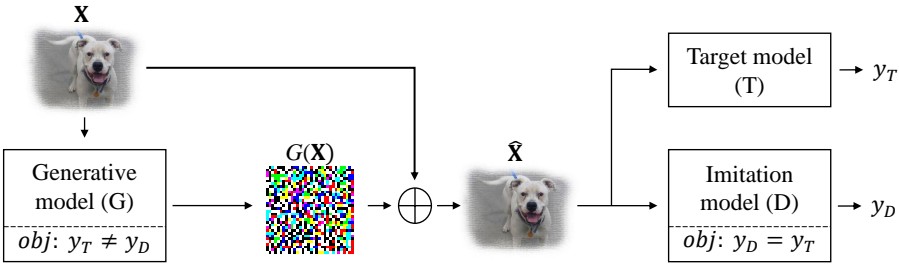

Figure 1: The proposed adversarial imitation attack. For the training stage, the objective of $G$ is to generate samples $\widehat{\mathbf{X}} = G(\mathbf{X}) + \mathbf{X}$ and let $y_D(\widehat{\mathbf{X}}) \neq y_T(\widehat{\mathbf{X}})$. The objective of $D$ is to guarantee $y_D(\widehat{\mathbf{X}}) = y_T(\widehat{\mathbf{X}})$. For the testing stage, the imitation model $D$ is utilized to generate adversarial examples to attack $T$.

$$\max_{G} \min_{D} \quad \mathcal{V}_{G,D} = -y_{T(\widehat{\mathbf{X}})}\log D(\widehat{\mathbf{X}}) - (1 - y_{T(\widehat{\mathbf{X}})})\log(1 - D(\widehat{\mathbf{X}}))$$
$$\text{subject to} \quad \widehat{\mathbf{X}} = G(\mathbf{X}) + \mathbf{X}.$$
(2)

Note that the $T$ is equivalent to the referee of this game. The two players $G$ and $D$ optimize their parameters based on the output $y_T(\widehat{\mathbf{X}})$. We suppose that adversarial perturbation $\epsilon \leq r_1$, and $\forall \widehat{\mathbf{X}} = G(\mathbf{X}) + \mathbf{X}, \|G(\mathbf{X})\| \leq r_2, r_1 \leq r_2$. If the $D$ can achieve $y_D(\widehat{\mathbf{X}}) = y_T(\widehat{\mathbf{X}})$, our imitation attack will have the same success rate as the white-box attack without the gradient information of $T$. Therefore, for a well-trained imitation network, adversarial examples generated by $D$ have strong transferability for $T$. A proper upper bound ($r_2 \geq r_1$) of $G(\mathbf{X})$ is the key points for training an efficient imitation network $D$. Especially in targeted attacks ($T$ outputs the specified wrong label), if the characteristics of $D$ is more similar to that of the attacked model, the transferability of adversarial examples will be stronger.

In the training stage, the loss function of $D$ is $\mathcal{J}_D = \mathcal{V}_{G,D}$. Because $G$ is more hard to train than $D$, sometimes the ability of $D$ is much stronger than $G$, so the loss of $G$ fluctuates during the training stage. In order to maintain the stability of training, the loss function of $G$ is designed as $\mathcal{J}_G = e^{-\mathcal{V}_{G,D}}$. Therefore, the global optimal imitation network $D$ is obtained if and only if $\forall \widehat{\mathbf{X}}, D(\widehat{\mathbf{X}}) = y_T(\widehat{\mathbf{X}})$. At this point, $\mathcal{J}_D = 0$ and $\mathcal{J}_G = e^0 = 1$. The loss of $G$ is always in a controllable value in training stage.

As we discussed before, if $r_1 \leq r_2$, the adversarial examples generated by a well-trained $D$ have strong transferability for $T$. Because the attack perturbation $\epsilon$ of adversarial examples is set to be a small value, we constrain the $\|G(\mathbf{X})\|$ in training stage to limit the search space of $G$, which can reduce the number of queries efficiently.

For training methodology, we alternately train the $G$ and $D$ in every mini-batch, and use $L_2$ penalty to constrain the search space of $G$. The procedure is shown in algorithm 1.

---

**Algorithm 1** Mini-batch stochastic gradient descent training of imitation network.

---

1 : **for** number of training iterations **do**
2 :     Sample mini-batch of $m$ examples$\{\mathbf{X}^{(1)}, \ldots, \mathbf{X}^{(m)}\}$ from training set.
3 :     Update the imitation model by descending its loss function :
4 :         $-y_T(\widehat{\mathbf{X}})\log D(\widehat{\mathbf{X}}) - (1 - y_T(\widehat{\mathbf{X}}))\log(1 - D(\widehat{\mathbf{X}}))$.
5 :     Update the generative model by descending its loss function :
6 :         $e^{-\mathcal{V}_{G,D}} + \alpha\|G(\widehat{\mathbf{X}})\|$.
7 : **end for**

---

In adversarial attacks, when the optimal $D$ is obtained, the adversarial examples generated by $D$ are utilized to attack $T$.

## 4 EXPERIMENTS

### 4.1 EXPERIMENT SETTING

In this subsection, we introduce the settings for our experiments.

**Datasets:** we evaluate our proposed method on MNIST (LeCun et al., 1998) and CIFAR-10 (Krizhevsky & Hinton, 2009). Because we need to use data different with the training set of $T$ to train the imitation network, we divided the test sets (10k) from MNIST and CIFAR-10 into two parts. One part contains 9500 images for training and another part contains 500 images for evaluating the attack performance.

**Model architecture and attack method:** in order to get the information of the attacked model $T$ as little as possible, we only utilize the output label (not the probability) of the $T$ to train the imitation network. The imitation network has no prior knowledge of the attacked model, which means it does not load any pre-trained model in experiments. For the experiments on MNIST, we design 3 different network architectures with different capacity (small network, medium network and large network) for evaluating the performance of our imitation attack with models having different capacity. We utilize the pre-trained medium network and VGG-16 (Simonyan & Zisserman, 2015) as the attacked model on MNIST and CIFAR-10, respectively. In order to compare the success rate of the proposed imitation attack with current substitute attack, we utilize 4 attack methods, FGSM, BIM, projected gradient descent (PGD) (Madry et al., 2018), C&W to generate adversarial examples. For testing, we use AdverTorch library (Ding et al., 2019) to generate adversarial examples. On the other hand, for comparing the performance of our method with current decision-based attacks and score-based attacks, we utilize Boundary Attack (Brendel et al., 2017), HSJA Attack (Chen et al., 2019), SimBA-DCT Attack (Guo et al., 2019) as comparison methods. Note that score-based attacks require output probabilities of $T$, which contain much more information than labels.

**Evaluation criteria:** the goals of non-targeted attack and targeted attack are to lead the attacked model to output a wrong label and a specific wrong label, respectively. In non-targeted attacks, we only generate adversarial examples on the images classified correctly by the attacked model. In targeted attacks, we only generate adversarial examples on the images which are not classified to the specific wrong labels. The success rates of adversarial attack are calculated by $n/m$, where $n$ and $m$ are the number of adversarial examples which can fool the attacked model and the total number of adversarial examples, respectively.

### 4.2 EXPERIMENTAL ANALYSIS OF ADVERSARIAL ATTACK

In this subsection, we utilize the proposed adversarial imitation training to train imitation models and evaluate the performance in terms of attack success rate.

To compare our method with substitute attack, We utilize the medium network and VGG-16 as attacked models on MNIST and CIFAR-10, respectively. Then we use the same train dataset to obtain a pre-trained large network (the architecture is also in Table 9) and ResNet-50 (He et al., 2016) to generate substitute adversarial examples. We obtain imitation networks by using the proposed adversarial imitation training. The large network and ResNet-16 are used as the model architectures of imitation networks on MNIST and CIFAR-10, respectively. The imitation models are only trained by 9500 samples of the test dataset, which are much less than the training sets of MNIST (60000 samples) and CIFAR-10 (50000 samples). The results of experiments are evaluated on the other 500 samples of the test dataset and are shown in Table 1 and Table 2. The success rates of the proposed imitation attack far exceed the success rates of the substitute attack in all experiments.

The experiments of Table 1 and Table 2 show that the proposed new attack mechanism needs less training images than substitute attacks, but achieves an attack success rate close to the white-box attack. The experiments indicate that the adversarial samples generated by a well-trained imitation model have a higher transferability than the substitute attack.

To compare our method with decision-based and score-based attacks, we evaluate the performance of these attacks. We utilize 9500 images from the test set of MNIST and CIFAR-10 to train the imitation network, and use other 500 images from the test set as unseen data for our method. The other decision-based and score-based attacks are evaluated on the test set of MNIST and CIFAR-10

Table 1: Performance of the proposed imitation attack compared with the white-box attacks and substitute attacks on MNIST. The architectures of networks are shown in Table 9. "White-box": generate adversarial examples from the attacked medium network. "Substitute": generate adversarial examples from the pre-trained large network. "Imitation": generate adversarial examples from the imitation large network. The numbers in ( ) denote the average $L_F$ perturbation distance per image.

| Attack | Non-targeted (%) | | | Targeted (%) | | |
|---|---|---|---|---|---|---|
| | White-box | Substitute | Imitation | White-box | Substitute | Imitation |
| FGSM | 82.90 (5.17) | 57.55 (5.17) | **75.45** (4.78) | 29.91 (5.25) | 15.81 (5.25) | **33.85** (5.17) |
| BIM | 86.52 (3.45) | 45.47 (3.45) | **70.62** (3.21) | 39.96 (3.45) | 14.03 (3.45) | **28.51** (3.53) |
| PGD | 65.79 (3.76) | 28.17 (3.76) | **49.70** (3.68) | 22.32 (3.84) | 7.80 (3.84) | **15.37** (3.84) |
| CW | 81.89 (3.06) | 38.63 (2.90) | **66.00** (3.14) | 43.08 (1.88) | 14.25 (1.56) | **37.32** (1.88) |

Table 2: Performance of the proposed imitation attack compared with the white-box attacks and substitute attacks on CIFAR-10. The attacked model is VGG-16.

| Attack | Non-targeted (%) | | | Targeted (%) | | |
|---|---|---|---|---|---|---|
| | White-box | Substitute | Imitation | White-box | Substitute | Imitation |
| FGSM | 84.87(2.73) | 39.29 (2.73) | **81.30** (2.73) | 41.42 (1.66) | 8.22 (1.66) | **32.65** (1.66) |
| BIM | 98.96 (1.08) | 47.27 (1.01) | **97.06** (1.14) | 67.82 (0.92) | 14.38 (0.83) | **62.56** (0.95) |
| PGD | 67.44 (1.11) | 30.25 (1.14) | **67.02** (1.29) | 28.17 (0.98) | 6.39 (1.01) | **25.57** (1.04) |
| CW | 97.69 (1.35) | 38.66 (1.35) | **70.40** (1.38) | 68.60 (1.51) | 20.78 (1.41) | **45.43** (1.51) |

dataset. Note that score-based attacks require much more information (output probabilities) than decision-based attacks (output labels). The results on MNIST and CIFAR-10 are shown in Table 3 and Table 4, respectively. Because our imitation attack only needs queries on the training stage, we evaluate performances of our method on its train and unseen sets. We set the iteration of adversarial imitation training to 1800, so the average number of query per image is 1800. We utilize BIM attack to generate adversarial examples as our imitation attack in this experiment.

This experiment shows that our imitation attack achieves state-of-the-art performance in decision-based methods. Even compared with the score-based attack, our imitation attack outperforms it in terms of perturbation distance and attack success rate. More importantly, it also obtains good results on unseen data, which indicates our imitation attack can be applied to query-independent scenarios.

### 4.3 EXPERIMENTAL ANALYSIS OF NETWORK CAPACITY

In the above subsection, we utilize a more complex network than the attacked model to replicate the attacked model. In this subsection, we study the impact of model capacity on the ability of imitation.

To evaluate the imitation performance of the network with less capacity than the attacked model, we train the small network, medium network, and large network to imitate the pre-trained medium network on MNIST dataset, and train VGG-13, VGG-16 and ResNet-50 to imitate VGG-16 on CIFAR-10. The performance of models with different capacities are shown in Table 5 and 6. The results show that an imitation model with a lower capacity than the attacked model can also achieve a good imitation performance. Attack success rates of all imitation models far exceed the substitute attacks in Table 1 and 2. Most experiments show that the larger capacity the imitation network has, the higher attack success rate it can achieve (FGSM, BIM, PGD in MNIST and BIM in CIFAR-10). However, some experiments show models having a larger capacity do not have a higher attack success rate (FGSM and PGD in CIFAR-10). We surmise that the performance of imitating an attacked model is not only influenced by the capacity of the imitation model $D$, but also influenced by the capability of the $G$.

### 4.4 EXPERIMENTAL ANALYSIS OF MODEL REPLICATION

In this subsection, we only use 200 images (20 samples per class) to train the imitation networks and discuss characteristics of the model replication.

Table 3: Performance of the proposed imitation attack compared with the decision-based and score-based attacks on MNIST. "Query": the average number of queries of attacks. "$L_F$": the average $L_F$ distance per image. "imitation-train": the performance of imitation attack on its training data. "imitation-unseen": the performance of imitation attack on other unseen data.

| Attack | Non-targeted | | | Targeted | | |
|---|---|---|---|---|---|---|
| | Query | $L_F$ | Success rate | Query | $L_F$ | Success rate |
| **Score-based** | | | | | | |
| SimBA | 752.32 | 5.41 | 96.78% | 879.86 | 3.53 | 82.63% |
| **Decision-based** | | | | | | |
| Boundary | 2830.44 | 9.49 | 100.0% | 2613.31 | 11.05 | 81.30% |
| HSJA | 4679.57 | 6.59 | 100.0% | 2113.06 | 6.27 | 75,67 |
| **Imitation-train** | **1800.00** | **4.94** | **99.62%** | **1800.00** | **5.33** | **84.36%** |
| **Imitation-unseen** | **—** | **4.94** | **99.52%** | **—** | **5.41** | **82.62%** |

Table 4: Performance of the proposed imitation attack compared with the decision-based and score-based attacks on CIFAR-10.

| Attack | Non-targeted | | | Targeted | | |
|---|---|---|---|---|---|---|
| | Query | $L_F$ | Success rate | Query | $L_F$ | Success rate |
| **Score-based** | | | | | | |
| SimBA | 501.65 | 1.76 | 99.37% | 489.90 | 2.06 | 84.16% |
| **Decision-based** | | | | | | |
| Boundary | 1937.02 | 3.26 | 100.0% | 1837.01 | 6.02 | 35.59% |
| HSJA | 2206.55 | 2.59 | 100.0% | 1527.84 | 1.12 | 32.53% |
| **Imitation-train** | **1800.00** | **1.29** | **99.87%** | **1800.00** | **1.19** | **87.76%** |
| **Imitation-unseen** | **—** | **1.29** | **99.58%** | **—** | **1.19** | **85.62%** |

Table 5: Performance of the imitation networks with different capacity on MNIST. The attacked model is pre-trained medium network. The architectures of networks are shown in Table 9. The adversarial examples generated by the same attack methods have the same perturbation distance.

| Attack | Non-targeted (%) | | | Targeted (%) | | |
|---|---|---|---|---|---|---|
| | Small net | Medium net | Large net | Small net | Medium net | Large net |
| FGSM | 69.82 | 63.98 | **75.45** | 33.18 | 28.06 | **33.85** |
| BIM | 61.17 | 57.75 | **70.62** | 21.36 | 18.71 | **28.51** |
| PGD | 42.45 | 39.84 | **49.70** | 10.45 | 10.47 | **91.67** |

We train the imitation network using 200 images from MNIST and CIFAR-10 test set, and compare its performance with Practical Attack (Papernot et al., 2017) on other images from MNIST and CIFAR-10 test set. The results are shown in Table 7 and 8. The practical attack uses the output labels of attacked models to train substitute models under the scenario, which they can make an infinite number of queries for attacked models. It is hard to generate adversarial examples to fool the attacked models by limited training samples. Note that what the substitute models imitate is the response for perturbations of the attacked model. A substitute model that can generate adversarial examples with a higher attack success rate is a better replica. Our adversarial imitation attack can produce a substitute model with much higher classification accuracy and attack success rate than Practical Attack for both non-targeted and targeted attacks in this scenario (with an infinite number of queries). We also show the performances of these two methods with limited query numbers in Figure 6. Additionally, the imitation model with low classification accuracy still can produce

Table 6: Performance of the imitation networks with different capacity on CIFAR-10. The attacked model is pre-trained VGG-16. The adversarial examples generated by the same attack methods have the same perturbation distance.

| Attack | Non-targeted (%) | | | Targeted (%) | | |
|---|---|---|---|---|---|---|
| | VGG-13 | VGG-16 | ResNet-50 | VGG-13 | VGG-16 | ResNet-50 |
| FGSM | 73.11 | **85.92** | 81.30 | 20.57 | 26.91 | **32.65** |
| BIM | 81.72 | 96.85 | **97.06** | 38.51 | 58.21 | **62.56** |
| PGD | 45.59 | **70.59** | 67.02 | 14.22 | **27.79** | 25.57 |

Table 7: Comparisons between imitation model and other substitute models. "Accuracy": the classification accuracy on other images from test set. The numbers in ( ) denote the average $L_F$ perturbation distance per image.

| | | | Non-targeted (%) | | |
|---|---|---|---|---|---|
| | Training data | Accuracy | BIM | FGSM | PGD |
| Practical | 200 (test set) | 80.43 | 14.08 (4.94) | 9.41 (5.10) | 2.63 (3.81) |
| Imitation | 200 (test set) | **97.68** | **71.03** (4.94) | **45.17** (5.12) | **14.81** (3.78) |

| | | | Targeted (%) | | |
|---|---|---|---|---|---|
| | Training data | Accuracy | BIM | FGSM | PGD |
| Practical | 200 (test set) | 80.43 | 2.51 (4.84) | 2.51 (5.14) | 1.11 (4.60) |
| Imitation | 200 (test set) | **97.68** | **26.65** (5.01) | **11.81** (5.23) | **15.66** (4.69) |

Table 8: Comparisons between imitation model and other substitute models. "Accuracy": the classification accuracy on other images from test set.

| | | | Non-targeted (%) | | |
|---|---|---|---|---|---|
| | Training data | Accuracy | BIM | FGSM | PGD |
| Practical | 200 (test set) | 29.83 | 2.31 (1.66) | 3.15 (1.65) | 2.73 (1.07) |
| Imitation | 200 (test set) | **71.09** | **64.50** (1.32) | **45.59** (1.65) | **20.80** (1.05) |

| | | | Targeted (%) | | |
|---|---|---|---|---|---|
| | Training data | Accuracy | BIM | FGSM | PGD |
| Practical | 200 (test set) | 29.83 | 1.14 (1.69) | 0.91 (1.65) | 0.68 (1.26) |
| Imitation | 200 (test set) | **71.09** | **23.06** (1.23) | **10.73** (1.65) | **12.10** (1.17) |

adversarial examples that have well transferability. It shows that our adversarial imitation training can efficiently steal the information from the attacked model of their classification characteristics under perturbations.

## 5 CONCLUSION

Practical adversarial attacks should have as little as possible knowledge of attacked model $T$. Current black-box attacks need numerous training images or queries to generate adversarial images. In this study, to address this problem, we combine the advantages of current black-box attacks and proposed a new attack mechanism, imitation attack, to replicate the information of the $T$, and generate adversarial examples fooling deep learning models efficiently. Compared with substitute attacks, imitation attack only requires much less data than the training set of $T$ and do not need the labels of the training data, but adversarial examples generated by imitation attack have stronger transferability for the $T$. Compared with score-based and decision-based attacks, our imitation attack only needs the same information with decision attacks, but achieves state-of-the-art performances and is query-independent on testing stage. Experiments showed the superiority of the proposed imitation attack. Additionally, we observed that deep learning classification model $T$ is easy to be stolen by limited unlabeled images, which are much fewer than the training images of $T$. In future work, we will evaluate the performance of the proposed adversarial imitation attack on other tasks except for image classification.

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

# A NETWORK ARCHITECTURES

Table 9: Network architectures for MNIST. Convolutional kernel $(A \times B, C)$ denotes the kernel size and channel number, respectively.

| ConvBlock | Small net | Medium net | Large net |
|---|---|---|---|
| ConvLayer $(A \times B, C)$ | ConvBlock $(5 \times 5, 20)$ | ConvBlock $(5 \times 5, 20)$ | ConvBlock $(5 \times 5, 20)$ |
| ReLU | ConvBlock $(5 \times 5, 50)$ | ConvBlock $(5 \times 5, 50)$ | ConvBlock $(5 \times 5, 50)$ |
| MaxPooling $(2 \times 2)$ | DenseLayer | ConvBlock $(3 \times 3, 50)$ | ConvBlock $(3 \times 3, 50)$ |
| | ReLU | DenseLayer | ConvBlock $(3 \times 3, 50)$ |
| | DenseLayer | ReLU | DenseLayer |
| | Sigmoid | DenseLayer | ReLU |
| | | Sigmoid | DenseLayer |
| | | | Sigmoid |

Table 10: Network architectures for generative model in our experiments.

| Layers | Parameter |
|---|---|
| ConvLayer | kernel: $3 \times 3$, stride: 1, padding:1, channel: 128 |
| BatchNorm | |
| LeakyReLU | |
| ConvLayer | kernel: $3 \times 3$, stride: 1, padding:1, channel: 512 |
| BatchNorm | |
| LeakyReLU | |
| ConvLayer | kernel: $3 \times 3$, stride: 1, padding:1, channel: 256 |
| BatchNorm | |
| LeakyReLU | |
| ConvLayer | kernel: $3 \times 3$, stride: 1, padding:1, channel: 128 |
| BatchNorm | |
| LeakyReLU | |
| ConvLayer | kernel: $3 \times 3$, stride: 1, padding:1, channel: 64 |
| BatchNorm | |
| LeakyReLU | |
| ConvLayer | kernel: $3 \times 3$, stride: 1, padding:1, channel: 3 |
| BatchNorm | |
| LeakyReLU | |

## B VISUALIZATION OF ADVERSARIAL EXAMPLES

In this section, we visualize the adversarial examples generated by the imitation model. The results are shown in Figure 2 and Figure 3. The experiments show that adversarial examples generated by the proposed imitation attack can fool the attacked model with a small perturbation.

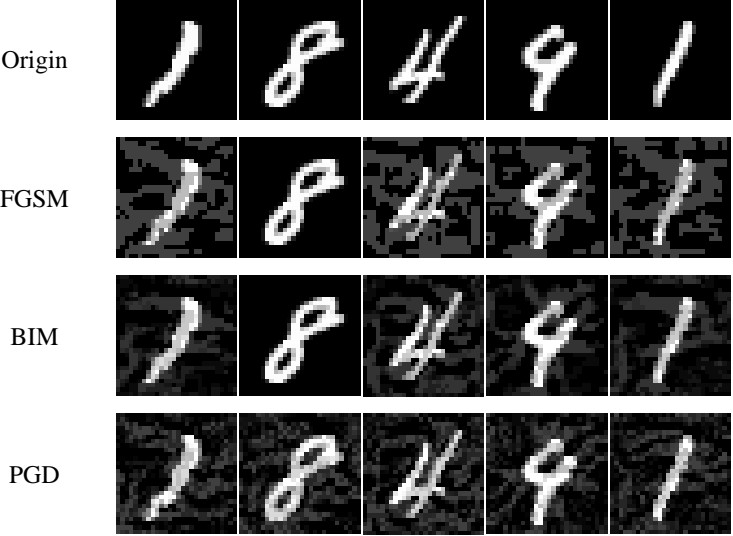

Figure 2: Visualization of some adversarial examples generated by the imitation model on MNIST. Original examples are in the first row. Examples from the second row to the third row are generated through FGSM, BIM, PGD, respectively.

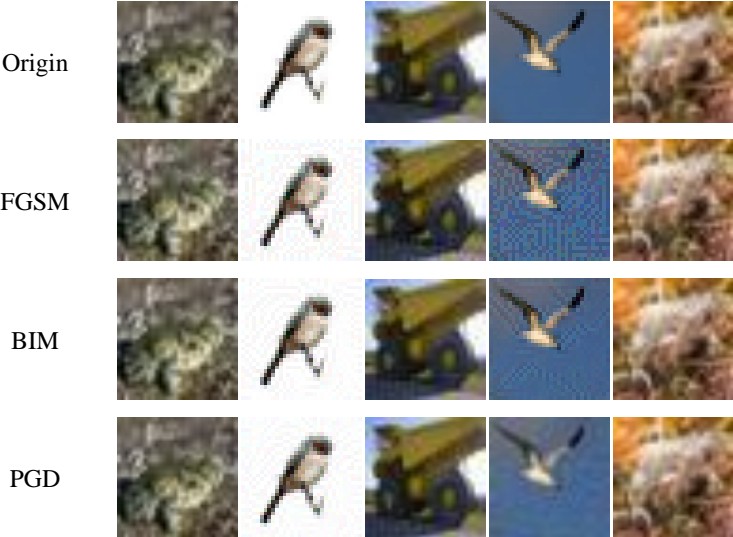

Figure 3: Visualization of some adversarial examples generated by the imitation model on CIFAR-10. Original examples are in the first row. Examples from the second row to the third row are generated through FGSM, BIM, PGD, respectively.

## C VISUALIZATION OF DISTURBANCES GENERATED BY GENERATOR IN TRAINING STAGE

In this section, we visualize the disturbances generated by generator in training stage.

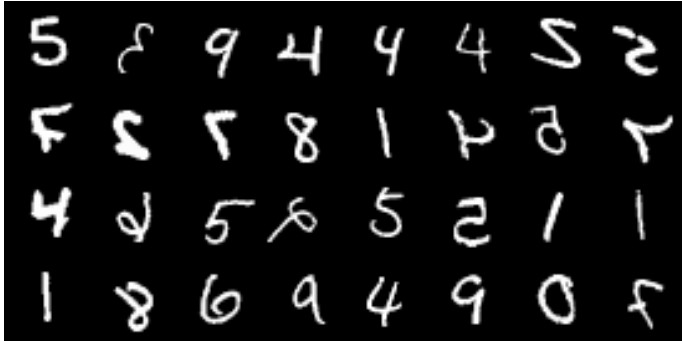

(a) Original images

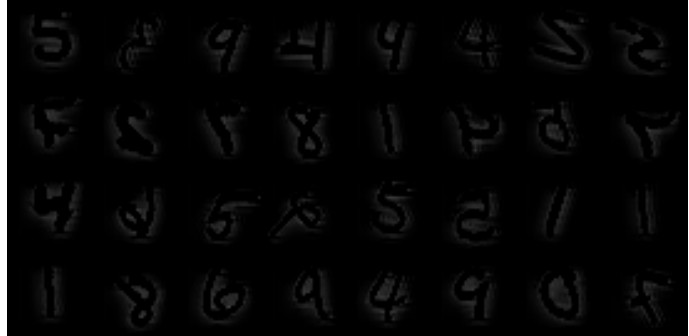

(b) Generated disturbances

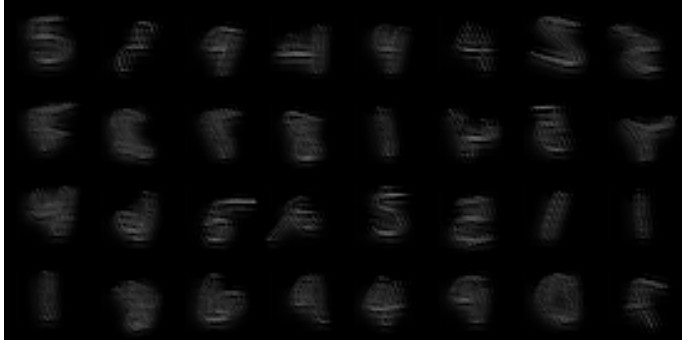

(c) Training samples for the imitation network

Figure 4: Visualization of some disturbances generated by the generator in training stage. The clean images added with the disturbances are used to train the imitation network.

# D ACCURACY CURVE OF THE IMITATION MODEL ON TRAINING STAGE

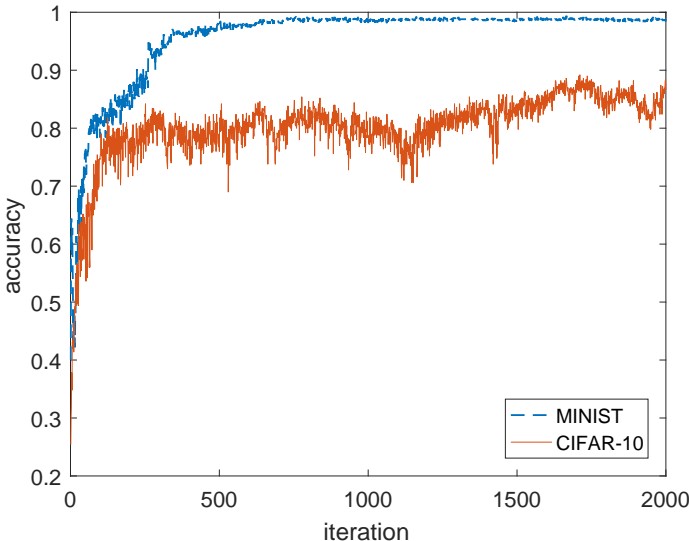

Figure 5: Accuracy curve of the imitation model on the 500 testing samples of MNIST and CIFAR-10 in adversarial imitation training. The number of training samples is 9500. We show the raw data without filtering.

# E  COMPARISON OF ATTACK SUCCESS RATE BETWEEN THE PROPOSED METHOD AND THE PRACTICAL METHOD

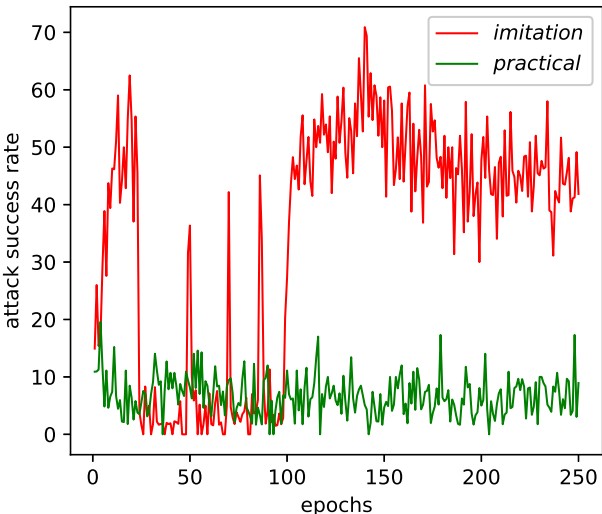

Figure 6: Comparison of attack success rate between the proposed method and the practical method on the early training stage.

