# OpenReview forum: "Adversarial Imitation Attack"
_ICLR.cc/2020/Conference — Reject_

### Official Review · AnonReviewer3 · 2019-10-21
**Official Blind Review #3**

**Rating:** 6

**Review:**

This paper proposes a new approach to conduct adversarial attacks, where an imitation classifier is trained to mimic the behaviours of the targeted/attacked classifier and adversarial attacks can be generated with existing attack methods on the imitation classifier. The training of the imitation classifier only requires the predictions of the targeted classifier, which may fit better in practice with limited access to the targeted classifier.

The reasons that I am going towards accept are as follows:

1. I feel that the idea of learning an imitation classifier is somehow novel and intuitive, which would improve the applicability of existing adversarial attacks in more realistic cases. In addition, using GAN framework in the training of the imitation classifier is also interesting.

2. The experiments are quite comprehensive and promising, including the comparisons with gradient-based attacks as well as the decision-based attacks. In addition, the different model configurations of the imitation classifier are also reported.

Suggestions:

1. It is a bit unclear of the model configuration of the generator, which seems to be not introduced in details.

2. It would be interesting to visualise what the generator generates.

Minor:

"sometimes the ability of G is much stronger than G"

**Experience Assessment:**

I have published one or two papers in this area.

**Review Assessment: Checking Correctness Of Derivations And Theory:**

I carefully checked the derivations and theory.

**Review Assessment: Checking Correctness Of Experiments:**

I assessed the sensibility of the experiments.

**Review Assessment: Thoroughness In Paper Reading:**

I read the paper thoroughly.

---

> ### Author Response · Authors · 2019-11-15
> **For your suggestions**
>
> Q1: ‘a bit unclear of the model configuration of the generator’
> A1: The model architecture of the generator is simple, and we found that the model configuration of the generator is not a key factor that influences training stability. In order to allow readers to get more information about our method, we add the model configuration of the generator in the latest version (see Table 10 on appendix A).
>
> Q2: ‘It would be interesting to visualize what the generator generates.’
> A2: According to this good suggestion, we visualize what the generator generates in the latest version (see Figure 4 on appendix C).
>
> Q3: ‘sometimes the ability of G is much stronger than G’
> A3: We have rewritten this sentence as ‘sometimes the ability of D is much stronger than G’.

---

### Official Review · AnonReviewer1 · 2019-10-23
**Official Blind Review #1**

**Rating:** 3

**Review:**

Authors propose a GAN-based adv imitation attack that can use less training data to produce a replica of the model.

The idea of using a GAN in producing adv examples in quite interesting. But the proposed approach is closely related to the following paper:

https://arxiv.org/pdf/1801.02610.pdf

Therefore, I am not sure about the novelty of the proposed approach.

Also can authors comment on the stability of GAN's training? Are any stabilizing methods integrated in GANs being used?

How does the proposed approach relate to the adversarial distillation literature?

**Experience Assessment:**

I have published in this field for several years.

**Review Assessment: Checking Correctness Of Derivations And Theory:**

I assessed the sensibility of the derivations and theory.

**Review Assessment: Checking Correctness Of Experiments:**

I assessed the sensibility of the experiments.

**Review Assessment: Thoroughness In Paper Reading:**

I made a quick assessment of this paper.

---

> ### Author Response · Authors · 2019-11-15
> **For the novelty and contribution**
>
> Q1: ‘the proposed approach is closely related to another paper’
> A1: Actually, our adversarial imitation attack is not like the method in that paper you mentioned. They developed a GAN to train a generator to generate adversarial examples, and accelerate adversarial training as defenses. However, we propose a new attack mechanism and develop a GAN, which is a special two players (contains a generator, an imitation model and an attacked model as a referee) game, to replicate the information of the attacked model. The role of the generator is to generate disturbances to explore the difference between the imitation model and the attacked model, but the generator in the paper you mentioned produces adversarial examples directly. The role of our imitation model is to learn the classification characteristics of the attacked model, using the data (labeled by the attacked model) with the disturbances, but the classifier in that paper is used to distinguish the generated data and the original instance. When the imitation model is obtained, adversarial examples are generated by this imitation model like current substitute attacks.
> The contributions of this paper are as follows:
> 1)	The proposed new attack mechanism needs less training data of attacked models than current substitute attacks but achieves an attack success rate close to the white-box attacks.
> 2)	The proposed new attack mechanism requires the same information of attacked models with decision attacks on the training stage but is query-independent on the testing stage.
>
> Q2: ‘Are any stabilizing methods integrated in GANs being used’
> A2: We use two tricks to stabilize the training of GANs. First, we constrain the norm of disturbances generated by the generator. Second, the loss of the generator has been carefully designed. These two tricks are introduced in the method section. In addition, the other parts are the same as conventional GAN.
>
> Q3: ‘How does the proposed approach relate to the adversarial distillation literature’
> A3: Our approach and adversarial distillation methods use adversarial training process including a generative model and classification model. But the details of the training and the purposes of these methods are very different. Our adversarial imitation attack utilizes GANs to train a replica of the attacked model, the adversarial distillation methods use GANs to train a model on other applications, such as adversarial defense. The form of GANs in adversarial distillation methods is like the original GANs, but our proposed method has 3 models including a generator, imitation model and attacked model. The attacked model is like a referee in this game and the other 2 players fight each other throughout the training. The roles of the generative model and the classifier are also different in these two kinds of methods.

---

### Official Review · AnonReviewer2 · 2019-10-24
**Official Blind Review #2**

**Rating:** 3

**Review:**

The authors propose to use a generative adversarial network to train a substitute that replicates (imitates) a learned model under attack. They then show that the adversarial examples for the substitute can be effectively used to attack the learned model. The idea is straightforward. The proposed approach leads to better success rates of attacking than other substitute-training approaches that require more training examples. Promising experimental results against decision and score-based attach schemes also demonstrate the effectiveness of the proposed approach.

My main concern is that the comparison to other approaches seem unfair, because

(1) Substitute models typically use all training data and query the learned model once per training example; the proposed approach uses fewer training data but needs 1800 queries per example. So it is not surprising that the proposed approach can be better than substitute models. As the authors focus on "practical" scenarios, the number of queries should be fixed as a constraint for all approaches to be fair.

(2) Having said (1), there is not enough detail in this paper to understand the similarity and difference between the proposed approach and decision-based ones (which is claimed to have similar query complexity during training). The authors mix the results with score-based (requiring more information), making the results more confusing to the readers. The authors are encouraged to present more detailed discussions on the comparison with decision-based competitors.

Also,

(3) One thing that is missing from the experiments is how well the replica clones the original model. All the information in the experiments are somewhat "indirect" (success rate, test accuracy, etc.) to answer this question, but there is no direct evidence. Is a well-cloned replica really a better substitute to construct adversarial examples? For instance, for replica with different "cloning accuracy" (rather than test accuracy), is a better replica also a better substitute? The paper fails to answer this question that best matches its design motivation.

The above are my main concerns. A minor one is

(4) The abstract that directly uses notations like T, G and D is horribly hard to read.


**Experience Assessment:**

I do not know much about this area.

**Review Assessment: Checking Correctness Of Derivations And Theory:**

I assessed the sensibility of the derivations and theory.

**Review Assessment: Checking Correctness Of Experiments:**

I assessed the sensibility of the experiments.

**Review Assessment: Thoroughness In Paper Reading:**

I read the paper at least twice and used my best judgement in assessing the paper.

---

> ### Author Response · Authors · 2019-11-15
> **For the experiment setting and explanation**
>
> Q1: ‘the number of queries should be fixed as a constraint for all approaches to be fair.’
> A1: Although we use more queries than the ‘practical’ method, we do not want to compare our method to other approaches in an unfair condition. This experiment shows that our method overcomes the difficulty of having little data (200). We choose the best model of ‘Practical’ method to evaluate its performance, but this performance cannot increase in additional queries.
> For this attack scenario. As the discussion in the paper of the ‘practical’ method, the main concern of this paper is to learn a substitute model using a small dataset, and they suppose their method could make an infinite number of queries for target models. We followed this scenario and proved that our method can succeed in a more extreme condition.
> However, it is true that we did not clearly describe this detail in the previous version. Other people may cannot clearly know the performance gap between the two methods under the same number of queries. To address these problems, we add the details of this scenario, to describe why we choose these results to compare in the current version. And we add a curve in appendix E in the current version, which shows the attack success rates of these two methods under the same limited number of queries (see change 1).
>
> Q2.1: ‘not enough detail to understand the similarity and difference between the proposed approach and decision-based ones’
> A2.1: Our proposed adversarial imitation attack is actually similar to the substitute attack but needs queries for target models to train a substitute attack. We discuss the similarity and difference between these two things in some parts of our paper, it is true that we need to integrate this information and emphasize it in one paragraph because we propose a new attack mechanism (see change 2).
>
> Q2.2: ‘mix the results with score-based’.
> A2.2: Because our method outperforms the current decision-based attacks, we want to further compare it with score-based attacks. Our methods have indeed surpassed them in some of the metrics. We hope it could provide additional information for who is interested in our method.
>
> Q3: ‘missing the detail how well the replica clones the original model’.
> A3: For we want to develop a method that can attack models efficiently, and the role of the generator is to add disturbances to the clean images, what the substitute models replicate is the classification characteristics of the attacked model under the perturbations. A substitute model that can generate adversarial examples with a higher attack success rate is a better replica. We explain this detail more clearly in the current paper (see change 1), so as not to cause confusion for readers.
>
> Q4: ‘abstract that directly uses notations like T, G and D is horribly hard to read’
> A4: For clearly explaining our idea, we have rewritten the abstract and avoid to use notations like ‘T, G, D’ in abstraction (see the abstract of the current version).
>
> ----------------------------
> Latest version changes
> ----------------------------
> Change 1:
> Section 4.3 on Page 6: ‘The practical attack uses the output labels of attacked models to train substitute models under the scenario, which they can make an infinite number of queries for attacked models. It is hard to generate adversarial examples to fool the attacked models by limited training samples. Note that what our adversarial imitation attack imitate is the response for perturbations of the attacked model. A substitute model that can generate adversarial examples with a higher attack success rate is a better replica. Our adversarial imitation attack can produce a substitute model with much higher classification accuracy and attack success rate than Practical Attack for both non-targeted and targeted attacks in this scenario (with an infinite number of queries). We also show the performances of these two methods with limited query numbers in Figure 5. Additionally, the imitation model with low classification accuracy still can produce adversarial examples that have well transferability. It shows that our adversarial imitation training can efficiently steal the information from the attacked model of their classification characteristics under perturbations’.
>
> Change 2:
> Section 1 on Page 2: ‘Score-based and decision-based attacks need a lot of queries to generate each adversarial attack. The similarity between the proposed method and current score-based and decision-based attacks is that adversarial imitation attack also needs to obtain a lot of queries in the training stage. The difference between them is our method does not need any additional queries in the test stage like other substitute attacks’.

---

### Decision · Program_Chairs · 2019-12-19

**Decision:**

Reject

**Comment:**

This paper proposes to use a generative adversarial network to train a substitute that replicates (imitates) a learned model under attack. It then shows that the adversarial examples for the substitute can be effectively used to attack the learned model. The proposed approach leads to better success rates of attacking than other substitute-training approaches that require more training examples. The condition to get a well-trained imitation model is that a sufficient number of queries are obtained from the target model. This paper has valuable contributions by developing an imitation attacker. However, some key issues remain. In particular, I agree with R1 that the average number of queries per image is relatively high, even during training. In the rebuttal, the authors made the assumption that “suppose their method could make an infinite number of queries for target models”, which is unfortunately not realistic. Another point that I found confusing: at testing, I don’t see how you can use the imitation model D to generate adversarial samples (D is a discriminative model, not a generator); it should be G, right?